# The Promises of Natural Killer Cell Therapy in Endometriosis

**DOI:** 10.3390/ijms23105539

**Published:** 2022-05-16

**Authors:** Janneke Hoogstad-van Evert, Romy Paap, Annemiek Nap, Renate van der Molen

**Affiliations:** 1Amphia Hospital, 4838 Breda, The Netherlands; 2Center of Translational Immunology, University Medical Center, 3553 Utrecht, The Netherlands; romypaap@ziggo.nl; 3Department of Obstetrics and Gynecology, Radboudumc, 6524 Nijmegen, The Netherlands; annemiek.nap@radboudumc.nl; 4Laboratory Sciences, Radboudumc, 6524 Nijmegen, The Netherlands; renate.vandermolen@radboudumc.nl

**Keywords:** endometriosis, natural killer cell, immunotherapy

## Abstract

Endometriosis is a gynaecological disease defined by the growth of endometrium-like tissue outside the uterus. The disease is present in approximately 5–10% of women of reproductive age and causes pelvic pain and infertility. The pathophysiology is not completely understood, but retrograde menstruation and deficiency in natural killer (NK) cells that clear endometriotic cells in the peritoneal cavity play an important role. Nowadays, hormonal therapy and surgery to remove endometriosis lesions are used as treatment. However, these therapies do not work for all patients, and hormonal therapy prevents patients from getting pregnant. Therefore, new treatment strategies should be developed. Since the cytotoxicity of NK cells is decreased in endometriosis, we performed a literature search into the possibility of NK cell therapy. Available treatment options include the inhibition of receptor–ligand interaction for KIR2DL1, NKG2A, LILRB1/2, and PD-1/PD-L1; inhibition of TGF-β; stimulation of NK cells with IL-2; and mycobacterial treatment with BCG. In preclinical work, these therapies show promising results but unfortunately have side effects, which have not specifically been studied in endometriosis patients. Before NK cell treatment can be used in the clinic, more research is needed.

## 1. Introduction

Endometriosis is a chronic disease that affects approximately 5–10% of women of reproductive age, which comes down to about 176 million women worldwide [1]. In women with endometriosis, endometrium-like tissue grows outside the uterus in the peritoneal cavity. This ectopic tissue may develop into endometriosis lesions if it adheres and invades into the peritoneum and thereafter obtains vascularization. Eutopic endometrium consists of stromal cells, epithelial cells, blood vessels, and leucocytes, which are important for embryo implantation and maintenance of pregnancy when present inside the uterus, but are of no use outside the uterus [1,2]. Endometriosis is most commonly found on the pelvic peritoneum, ovaries, and rectovaginal septum. Patients with endometriosis suffer from dysmenorrhoea, dyspareunia, chronic (non-menstrual) pelvic pain, irregular uterine bleeding, and infertility. The condition is present in 35–50% of women suffering from pain, infertility, or both [3]. Women with endometriosis have a higher risk of emergency department visits and hospitalization and significant physical and social constraints. Thereby, endometriosis is associated with reduced productivity, increased absenteeism from school and work, and an overall reduced quality of life [4,5]. 

There are three different phenotypes of endometriosis: superficial peritoneal endometriosis, deeply infiltrating endometriosis (DIE), and endometriomas or ovarian cystic endometriosis. In superficial peritoneal endometriosis, lesions are mostly present on the peritoneum. To be classified as DIE, a lesion should extend more than 5 mm beneath the peritoneum [6,7]. Endometriomas are ovarian cysts with a wall often consisting of endometrium-like or fibrotic tissue and are filled with blood products [8]. The pathophysiology of endometriosis is not fully known yet. The origin of the disease is presumably multifactorial, and different theories exist explaining the pathogenesis. These theories can be divided into two categories: those that claim lesions arise from the uterine endometrium and those that hypothesize lesions originate from tissues other than the uterus. The most common hypothesis is the one of retrograde menstruation. 

In order to develop endometriosis, endometrium-like tissue has to reach the peritoneal cavity. However, in almost all women with patent tubes, endometrial cells are present in the peritoneal cavity due to retrograde menstruation, but only a small fraction of these women develop endometriosis. Therefore, we can assume that other factors are important in the development of endometriosis such as cell survival, angiogenesis, cell growth, and immune response [9,10,11]. In healthy women, cells that reach the peritoneal cavity are eliminated by immunosurveillance and apoptosis. In women with endometriosis, changes in the cell-mediated and humoral immunity result in reduced clearance of endometrial cells and, therefore, implantation and development of endometriosis lesions [10]. There are several hypotheses on how the cells can escape immunosurveillance.

Endometrial cells in the peritoneal cavity overexpress anti-apoptotic factors and show a decreased expression of proapoptotic factors [12]. Besides this, the Fas–FasL apoptosis pathway plays an important role in immunosurveillance in the peritoneal cavity. On stromal cells present in an endometriosis environment, the FasL expression is increased, leading to Fas-mediated apoptosis for immune cells that express Fas on their membrane such as T cells and natural killer (NK) cells leading to their escape from immunosurveillance [13]. Macrophages are the most prevalent type of immune cells found in the peritoneal cavity. In the physiological situation, macrophages remove blood cells, damaged tissue, and cellular debris from the peritoneal cavity by phagocytosis. Although the number and activation of macrophages is upregulated in endometriosis, the phagocytic activity is reduced, resulting in failure to eliminate ectopic endometrial cells that end up in the peritoneal cavity [10,11]. Dendritic cells (DCs) are responsible for antigen presentation and thereby initiate the response of the adaptive immune system. Whether DCs also play a role in the angiogenesis of endometriosis lesions is not fully clarified [14,15,16]. Different subsets of T cells play a role in the pathogenesis of endometriosis. In endometriosis lesions, there is a lower frequency of pro-inflammatory Th1 cells, a subset of CD4+ T cells, compared to that in the eutopic endometrium. Anti-inflammatory Th2 cytokines such as interleukin 10 (IL-10) and IL-4 are increased, while Th1 cytokines such as interferon gamma (IFN-γ) are decreased. The difference in Th1/Th2 balance between the local lesions and peripheral blood has not been explained yet [10,11]. Regulatory T cells (Treg) were found to be increased in circulation in endometriosis patients, but not in the eutopic nor ectopic lesions; this may be considered as a compensatory mechanism to regulate the inflammatory condition in this disease [17,18]. Furthermore, a higher CD4/CD8 ratio and an increased concentration of each subset was found in the peritoneal fluid of patients with endometriosis. The concentration of T cells was higher in endometriosis lesions compared to that in eutopic endometrium, but the CD4/CD8 ratio was similar [10,11]. 

NK cells are suggested to play an important role in the pathogenesis of endometriosis. They are important in killing malignant and infected cells and have roles in tissue remodelling in different organs such as the uterus. In the healthy situation, NK cells make up approximately 15% of all circulating lymphocytes [1,19]. The function of NK cells is controlled by signals that are initiated by different combinations of activating or inhibiting molecules found on the surface of the cells [1]. NK cells kill other cells by secreting lytic granules that contain granzymes, perforin, and cytotoxins or cytokines such as IFN-γ [19]. See Figure 1. Based on the level of CD56 expressed on the surface of NK cells, two subcategories can be distinguished. CD56^dim^ NK cells express less CD56 and have higher cytotoxicity, secrete less cytokines, and have higher levels of the immunoglobin-like NK cell receptors and FC-γ receptor III (CD16). CD56^bright^ NK cells express a high level of CD56 and have low cytotoxicity but are potent producers of cytokines such as IFN-γ and TNF-α and can thereby activate monocytes such as macrophages. CD56^bright^ NK cells have low or absent levels of the FC-γ receptor CD16 and are important for immune regulation [1,19]. In the circulation, most of the NK cells present are CD56^dim^ while in other tissue such as the endometrium mostly CD56^bright^ NK cells are present. NK cells in the uterus are called uterine NK cells (uNK) and are CD56^superbright^ [1,20]. In the menstrual cycle, the percentage of uNK cells in leucocytes fluctuates between 35% in the proliferative phase and 70% in the late secretory phase [21]. These uNK cells were significantly higher in the eutopic endometria of women with endometriosis than in the endometria of normal controls [22]. uNK cells play an important role in pregnancy and trophoblast invasion [23,24]. 

In endometriosis, the phenotype and function of NK cells differs from that of healthy women. Peripheral blood NK cells (pNKs) are observed to have an increased expression of killer cell inhibitory receptors (KIRs). These receptors contain an Ig domain (KIR2DL1, KIR2DL2, KIR3DL1, and KIR3DL2) extracellularly and immunoreceptor tyrosine-based inhibitory motifs (ITIMs) intracellularly.

The binding of a ligand to the receptor leads to immune suppression including suppression of NK cell cytotoxicity. Levels of ITIM-KIRs, KIR2DL1, and intracellular adhesion molecule-1 (I-CAM) are upregulated on pNK cells of patients with endometriosis, causing inhibition in pNK cell cytotoxicity [1,19,25,26]. This could result in the low success in clearance of endometrial cells in the peritoneal cavity [1]. In women with endometriosis, an increased number of immature pNK cells is found. Interestingly, removal of lesions by surgery induces an increase in the proportion of mature pNK cells. This suggests that lesions have an effect on pNK cell development by the production of certain cytokines [1,19]. When investigating local resident NK cells, peritoneal fluid is an interesting source. In peritoneal NK cells of endometriosis patients, the amount of cytotoxic NK cell markers is reduced as well as the cytotoxicity markers on the available NK cells [19]. Thereby, the overall activity of peritoneal NK cells is decreased. Furthermore, the levels of granzyme B, perforin, TRAIL, and CD107a are reduced in the peritoneal fluid of patients with endometriosis indicating that the NK cells are functionally defective [19].

Up till now, NK cell therapy has mainly been used to treat haematological cancers, but its effectivity is also being investigated for solid tumours. NK cells are thought to react to tumour cells because of the “missing-self hypothesis”. NK cells monitor and recognize self-major histocompatibility complex 1 (MHC-I), which are molecules present on normal cells. When MHC-I is present on the cell surface, the function of NK cells is inhibited, and the cell is not killed [27]. With adoptive NK cell therapy, NK cells are isolated, stimulated and expanded ex vivo and administered to the patient intravenously. In order to stimulate the NK cells, cytokines such as IL-2, IL-15, IL-12, and IL-18 or feeder cells need to be added [27,28,29]. There are two types of adoptive NK cell therapy: autologous and allogeneic. Allogeneic NK cells can be derived from umbilical cord blood, cell lines such as NK92, induced pluripotent stem cells (iPSCs), and adult donor lymphapheresis products [28,29]. Some promising results are available in haematological cancers, and numerous trials are exploring the safety and efficacy of NK cell adoptive transfer in solid tumours.

Different strategies are being explored to improve NK cell therapy: cytokine-based therapy stimulates NK cells with different cytokines such as IL-2, IL-15, IL-12, and IL-18, whereas the administration of IL-2 causes an increase in NK cell proliferation and cytotoxicity [30]. Unfortunately, IL-2 also increases the expansion of Treg cells, which may suppress NK cell activity [27,28]. IL-15 significantly activates NK cells, increases their cytotoxic activity, and has no effect on Treg cells. To improve antibody-dependent cytotoxicity, NK cell function can be improved by blocking inhibitory receptors or engaging activating receptors. For instance, KIR receptors can be blocked using Lirilumab or the activating receptor SLAMF7 on NK cells can be stimulated using Elotuzumab [27]. Another option is to genetically modify NK cells to increase their effectiveness, for instance, with chimeric antigen receptor (CAR)–engineered NK cells. CAR-NKs are synthetic receptors that can bind a specific antigen. When the CAR-NK receptor and antigen interact, cytotoxicity is triggered against tumour-representing antigens leading to lysis [27]. NK cell adoptive therapy is often administered intravenously but can also be administered intraperitoneally in case of peritoneal disease [28]. In endometriosis, intraperitoneal NK cell therapy might be an option. Since approximately 30% of women still experience pain when receiving hormonal or surgical treatment [31], there is an urgent need to find other treatment options. Since the function of NK cells is decreased in women with endometriosis, the purpose of this review is to investigate whether NK cell therapy would be beneficial in endometriosis patients and to investigate what the current status of NK cell therapy is.

## 2. Results

From the ten selected articles, we collected evidence concerning NK cell immunology in endometriosis and on which inhibitory receptor–ligand interactions are increased. The most important inhibitory NK cell receptors mentioned as possible checkpoints for the elimination of ectopic endometrial cells are KIR2DL1, leukocyte immunoglobulin-like receptors (LILRs), and CD94/NKG2A [32]. Below, different studies are described that have investigated the role of these receptors in endometriosis. All data were obtained in animal models except when stated otherwise.

### 2.1. Blocking of Inhibitory Receptors

Matsuoka et al. described in 2005 that KIR2DL1, an inhibitory receptor, is increased on NK cells of women with pelvic endometriosis. In this study, 24 Japanese women with endometriosis and 25 women with other conditions confirmed by laparoscopy were included. Peripheral blood (PB) and peritoneal fluid (PF) samples were taken and analysed by flow cytometry and Western blot. No significant differences were found in CD56+ NK cell ratio in PBMCs, but the percentage of KIR2DL1-positive NK cells among NK cells in both the PB and PF was significantly higher in endometriosis patients [33].

Binding of human leukocyte antigen (HLA) G to its receptors LILRB1 and LILRB2 on NK cells is important in the pathogenesis of endometriosis. When HLA-G, which is expressed on ectopic endometrial tissue, binds the receptors, NK cell function is inhibited. In blood samples from 276 endometriosis patients and 314 controls, Bylinska et al. investigated whether polymorphisms in HLA-G, KIR2DL4, LILRB1, and LILRB2 influence the susceptibility to endometriosis. This was done by PCR, PCR-restriction fragment length polymorphism (PCR-RFLP), and allelic discrimination methods. Certain polymorphisms of HLA-G protected against endometriosis, and polymorphisms of LILRB1 and LILRB2 were associated with susceptibility to endometriosis and its progression. The most important conclusion was that HLA-G and the receptors LILRB1 and LILRB2 may play a crucial role in eliminating endometriotic cells and the development of endometriosis [34].

Patients with endometriosis have a higher expression of the CD94/NKG2A receptor on NK cells, and upon binding of its ligand HLA-E, NK cell function is inhibited. The study of Galandrini et al. compared PB, PF, endometriotic tissue, and endometrial tissue samples of 20 women with confirmed endometriosis to the samples of 12 controls. By flow cytometry, reverse-transcription PCR, and a cytotoxicity assay, the authors showed that a significantly higher proportion of PF NK cells in the endometriosis group expressed CD94/NKG2A compared to the control group. Furthermore, transcription of the HLA-E gene is high in endometriotic tissue, resulting in a high HLA-E mRNA level. After cloning NK cells, which were isolated from the PF by negative selection, the cytotoxic activity of CD94/NKG2A-positive and -negative NK cells toward the HLA-E-expressing DT360 cell line was investigated. CD94/NKG2A-positive NK cells could not kill DT360 cells, while CD94/NKG2A-negative clones could. When masking CD94/NKG2A receptors with anti-NKG2A antibodies, lysis of the NK cells was restored and comparable to the lysis of HLA-E-negative 721.221 cells. From these results, it can be concluded that the CD94/NKG2A receptor on PF NK cells is functionally active and that the cytotoxic function is inhibited upon interaction with HLA-E [35]. 

### 2.2. Checkpoint Inhibition

Checkpoint inhibition with anti-PD-1/anti-PD-L1 therapy could also be a treatment for endometriosis [32]. Wu et al. found in 2018 that the PD-1/PD-L1 expression in the eutopic and ectopic endometrium was increased in endometriosis patients. Blood and eutopic and ectopic endometria from 15 endometriosis patients and 15 controls were tested for PD-1/PD-L1 expression by immunohistochemistry, Western blot, and flow cytometry. When PD-1 on NK cell binds to PD-L1 on endometriosis cells, NK cell function is inhibited [36]. See Figure 2.

### 2.3. Inhibitory Cytokine Therapy

The role of TGF-β to inhibit factors that could be involved in the decreased NK cell cytotoxicity was explored in a study with 51 patients with confirmed endometriosis and 15 controls (Sikora et al.). A cytokine assay was performed to examine the levels of TGF-β1, TGF-β2, TGF-β3, and other cytokines. The levels of all TGF-β isoforms were higher in women with endometriosis in both PF and serum. Moreover, the levels of the pro-inflammatory cytokines IL-1, IL-6, and IL-17AF, and the anti-inflammatory cytokine IL-10, were increased in both PF and serum [37]. Zhang et al. observed that both the expression levels of angiotensin receptor 1 (AGTR1) and the activity of NF-κβ were increased in human endometriosis tissues and stromal cells. Moreover, oestrogen was found to regulate the expression levels of AGTR1 in stromal cells, thereby, possibly contributing to the pathogenesis of endometriosis [38]. Unfortunately, there is no data available in humans on the blocking of angiotensin receptors. In an endometriosis rat model, the angiotensin II receptor blocker losartan reduced TGF-β, and the surface area of endometriosis lesions was decreased as were the plasma levels of the angiogenic factors and inflammatory markers VEGF, TNF-α, PTX-3, and CRP [39,40]. Sprague–Dawley albino rats underwent surgery to implant autologous endometrial tissue onto the peritoneal surface. The rats were divided into two groups: eight rats received oral treatment with water and eight rats received oral treatment with losartan. After four weeks, the size of the implants was determined as were the plasma cytokine levels. The surface of the implant and the extent and severity of adhesions were significantly decreased in the losartan group as were the VEGF, TNF-α, PTX-3, and CRP plasma levels [39].

### 2.4. Stimulatory Cytokine Therapy

Besides anti-inhibitory cytokine therapy, stimulatory cytokine therapy with IL-2 could be a treatment option [32]. Again, no human data are available. Velasco et al. showed in 2006 in a rat model with endometriosis that treatment with IL-2 caused a reduction in endometriosis lesion size. In this in vivo study, 20 female Wistar rats underwent surgery to implant autologous endometrium tissue onto the inner surface of the abdominal wall, and 20 rats underwent sham surgery. After one month, the rats were treated with IL-2 or placebo intraperitoneally and one month later underwent surgery to determine the implant size and received a second dose of treatment. Blood samples and PF were collected at the first surgery and before sacrifice. After IL-2 treatment, the rats showed a significant reduction in implant size compared to initial size after the third and last measurement. At the last measurement, the size was also significantly different from the placebo group. Furthermore, peritoneal NK cells and DCs were enhanced in rats with endometriosis treated with IL-2, and a greater number of activated lymphocytes, macrophages, NK cells, and DCs was observed in the implants [41].

### 2.5. Mycobacterial Treatment

Treatment with mycobacteria in endometriosis causes effective killing of endometrial stromal cells. In the study by Clayton et al., endometrial stromal cells were obtained from women with endometriosis, and PBMCs were obtained from healthy female donors from which NK cells were isolated using CD56-labelled MACS beads. Different multiplicities of infection (m.o.i.) of two different strains Bacillus of Calmette and Guérin (BCG), BCG Pasteur and BCG Connaught, were used as treatment to stimulate NK cells. [42]. Clayton et al. found that PBMCs stimulated with 0.1 m.o.i. of BCG Connaught for seven days, showed an increased cytotoxicity by ^51^Cr release compared to untreated PBMCs. When NK cells were depleted from the treated PBMCs, no change in ^51^Cr release was observed compared to the untreated PBMCs. NK cells isolated from the BCG-treated PBMCs caused the highest increase in ^51^Cr release, meaning that BCG treatment mostly affects NK cell cytotoxicity and treated NK cells are more effective in killing endometrial stromal cells.

### 2.6. Adoptive NK Cell Therapy

Recently, the first study into autologous NK cell therapy in endometriosis patients was started. At the Shenzhen People’s Hospital in China, 60 women diagnosed with stage III–IV endometriosis are taking part in this phase 1 trial. Half of them will receive standard care with a gonadotrophin-releasing hormone agonist (GnRH agonist) combined with reverse addition therapy, and the other half will receive the same therapy combined with autologous NK cell therapy. The outcome measures will be incidence of treatment-emergent adverse events, purity and function of NK cells, endocrine hormone levels, pain score, and percent of pregnancy (NCT03948828).

## 3. Discussion

In this review, the current status of NK cell therapy research in endometriosis is summarized. NK cells have a prominent role in the pathogenesis of endometriosis, but research in humans describing the possibilities of NK cell therapy to reduce the symptoms of this disease is limited. Although more research should be performed with regard to the use of NK cell therapy in endometriosis, promising results are available of possible targets for NK cell therapy. With these data, there is a basis for further research into the application of NK cell adoptive therapy and for the development of strategies to improve NK cell function in endometriosis patients. 

Most data in this review were obtained from animal models. These models are useful to check how in vitro experiments work out in a microenvironment that resembles the situation in humans. However, there are limitations on animal models in endometriosis. The validity of endometrium tissue in the peritoneal cavity in predicting effect in human endometriosis can be debated. Moreover, the peritoneal cavity is a very specific microenvironment for many aspects such as cytokines, immune cells, microbioma, and steroid hormones. In this difficult-to-mimic environment, we tried to place the results in a frame, realizing that more research in humans is essential to find new possible therapies for endometriosis. 

Research into NK cell stimulating therapies for endometriosis can be divided into four categories: (1) blocking of inhibitory receptors such as KIR2DL1, LILRB1/2, NKG2A and PD-1/PD-L1; (2) inhibiting cytokines such as TGF-β; (3) stimulating NK cell with IL-2, and (4) mycobacterial treatment with BCG. Besides, research is being done on the use of adoptive NK cell therapy in endometriosis. In Figure 1, an overview is provided about NK cell therapy options in endometriosis.

The inhibitory receptor KIR2DL1 is upregulated on both peripheral blood and peritoneal NK cells in endometriosis patients. Upon binding of its ligand HLA-C2, this ITIM-KIR inhibits NK cell cytotoxicity [32,33]. Binyamin et al. observed that blocking KIR2DL1 on NK cells of a healthy donor expressing a high level of KIR2DL1 resulted in an increase in NK cell cytotoxicity against B cells. Blocking KIR2DL1 combined with the anti-CD20 monoclonal antibody rituximab resulted in a higher cytotoxicity compared to rituximab alone. This shows that blocking KIR1DL2 has a positive effect on NK cells but may be not effective enough on its own [43]. Blocking the interaction between KIR2DL1 and its ligand could be a possible treatment in endometriosis but should be examined further on endometriosis cells and lesions before conclusions can be drawn.

Since Bylinska et al. demonstrated the importance of HLA-G binding to LILRB1/LILRB2 in the pathogenesis of endometriosis; inhibiting this interaction might be a good treatment strategy [34]. However, HLA-G is of value in protecting the foetus from destruction by the immune system of the mother [44]. Therefore, blocking the interaction between LILRB1/LILRB2 while the patient is trying to get pregnant, will not be an option. Furthermore, HLA-G is important in organizing immune responses and maintaining the tolerance in inflammatory and autoimmune diseases. Because of this, blocking the interaction with its receptors probably has side effects [45]. To our knowledge, no treatment exists that disturbs the interaction between this ligand and receptors.

CD94/NKG2A is upregulated in the peritoneal fluid of endometriosis patients and decreases NK cell cytotoxicity upon binding of its ligand HLA-E. Inhibiting this interaction results in an increase in NK cell reactivity, as shown by Galandrini et al. [35]. In 2018, André et al. reported that NKG2A targeting with monalizumab enhanced NK cell activity against tumour cells [46]. A phase II trial with monalizumab plus the EGFR inhibitor cetuximab in 40 patients with squamous cell carcinoma of the head and neck resulted in a response rate of 31%. The most common reported side effects due to monalizumab were fatigue (17%), pyrexia (13%), and headache (10%) [46]. Monalizumab is also studied in gynaecologic malignancies. The response rate to monalizumab was 39% in a phase II trial that included 58 pre-treated patients. The most common adverse events were fatigue, headache, and vomiting reported in more than 20% of the patients [47]. Since this treatment is effective in cancer, NK cell therapy with anti-NKG2A could be a promising therapy in endometriosis although the efficacy should be increased, and the treatment has some side effects.

Because of the increased expression of PD-1/PD-L1 in endometriosis tissue, blocking this interaction could be interesting as an endometriosis treatment [36]. Nowadays, anti-PD-1/PD-L1 therapy is effectively used in the treatment of, amongst others, non-small cell lung cancer, classical Hodgkin lymphoma, melanoma, and renal cell carcinoma [48]. Treatment with anti-PD1-1/PD-L1 can give rise to adverse events such as diarrhoea, abdominal pain, immune-related hepatitis, or other autoimmune-related features such as rash, arthralgia, and myalgia [49]. More research is needed before PD-1/PD-L1 treatment can be implemented in endometriosis since the mechanism of PD-1/PD-L1 regulation in this disease is not fully understood yet [36]. 

As Sikora et al. showed, TGF-β is upregulated in women with endometriosis and inhibits the function of NK cells. Furthermore, TGF-β also inhibits the function cytotoxic T cell, DCs, macrophages, and neutrophils and induces Treg and Th17 differentiation. Because of this, it is thought that TGF-β is important in the pathogenesis of endometriosis [37]. Losartan can reduce the size of endometriotic lesions as was shown in a rat model. However, the main goal of losartan is not to reduce TGF-β but to block the angiotensin II receptor in hypertension patients, angiotensin II induces TGF-β. It cannot be concluded from the study by Cakmak et al. that TGF-β reduces endometriotic lesions, especially because there are no studies in humans [39]. Several anti-TGF-β therapies are being developed and tested in clinical trials. Because TGF-β is expressed all over the body by almost all cell types and is involved in many physiological processes such as angiogenesis, cell proliferation, and differentiation, systemic inhibition could have severe side effects. A study to the anti-tumour effects in mice showed that a long-term effect of anti-TGF-β therapy with LY2109761 is the growth of aggressive carcinomas non-responsive to this drug. However, this is only one of the drugs developed to inhibit TGF-β, and other therapies show more promising results in the treatment of cancer [50]. Anti-TGF-β therapy is studied in other diseases such as systemic sclerosis, skin fibrosis, and diabetes [51]. A few of the currently used anti-TGF-β drugs are statins for hypercholesterolaemia and endothelin-1 blockers for pulmonary hypertension [52]. These drugs result in partial abrogation of TGF-β activity, with only modest toxicity in animal models. Despite the potential side effects, anti-TGF-β therapy shows promising results. Therefore, it might be interesting to look further into this treatment as an option for endometriosis when the efficacy is higher and the long-term effects are known.

Stimulation of NK cell by IL-2-generated lymphokine-activated killer (LAK) cells, which shows an increase in cytotoxic potential against tumour cells. Therefore, IL-2 therapy is used as treatment in metastatic renal cell carcinoma and metastatic melanoma [53]. LAK cells were also found to have an increased cytotoxicity against endometriotic cells [54]. However, as mentioned in the introduction, IL-2 also has a stimulating effect on Tregs. Because of the positive results in cancer and the results from the study by Velasco et al., IL-2 as treatment for endometriosis should be looked into further.

Although BCG treatment seems to be promising in stimulating NK cells, Clayton et al. point out that the treatment, as it is currently used in cancer treatment, is unlikely to be acceptable as treatment in endometriosis because of the small mortality risk from BCG sepsis. Therefore, mycobacterial therapy for endometriosis is not the most promising option [42].

Taken together, these findings show interesting approaches for new treatment options for endometriosis. However, none of the optional treatment strategies described are being studied specifically for endometriosis in clinical trials and most of them have serious side effects in animal models. Only IL-2 and losartan treatment have been tested in an endometriosis rat model with promising results. Before we can conclude whether NK cell therapy is an option for endometriosis, more research should be performed into the role of NK cells in the pathogenesis of endometriosis, and additional clinical trials targeting NK cells should be performed.

## 4. Materials and Methods

In order to answer our question, a search in PubMed was performed in December 2021, using a combination of the search terms: NK cell therapy, natural killer cell therapy, immunotherapy, immune therapy, and endometriosis. With these terms, 26 articles were found from 2000 to 2021, and screened for relevance by title and abstract. By cross references, relevant articles were selected. The articles were screened, and the reasons for exclusion were as follows: articles not on endometriosis or not on immunotherapy, reviews, and description of murine model development. Ten articles were included for the narrative review. To find current and future trials on NK cell therapy or immunotherapy and endometriosis, a search was performed at clinicaltrials.gov. Five trials were found of which one was relevant and included. Other trials did not include immunotherapeutic treatment of endometriosis.

## Figures and Tables

**Figure 1 ijms-23-05539-f001:**
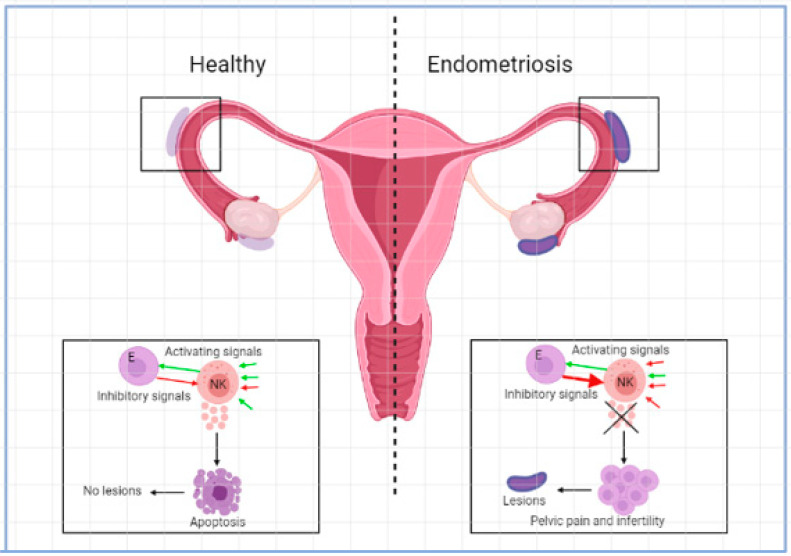
Activating and inhibitory signals in NK cell microenvironment in the development of endometriosis.

**Figure 2 ijms-23-05539-f002:**
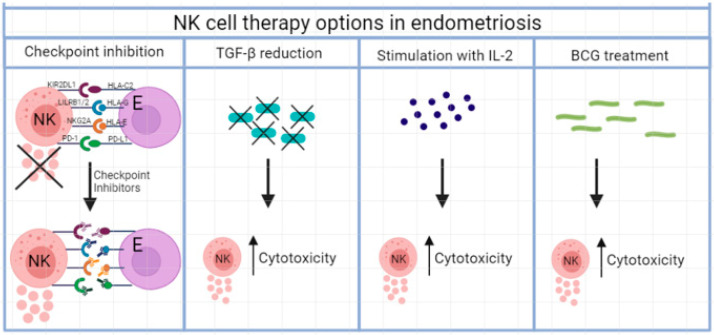
Different NK cell therapy options in endometriosis, whereby cytokine release is improved and NK cell cytotoxicity increased.

## Data Availability

Not applicable.

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
