# Peer review of "The Promises of Natural Killer Cell Therapy in Endometriosis"

_ijms, 2022, doi:10.3390/ijms23105539_

Round 1

Reviewer 1 Report

The manuscript ‘ The Promises of natural killer cell therapy in endometriosis’ reviews NK cell therapy as a new approach to treat endometriosis.

The topic is interesting and data seem exhaustive, but the manuscript is difficult to read, is poorly structured, and reads more like a compilation than a review.  Only in the discussion is found  ‘In this review the current status of NK cell therapy research in endometriosis is summarized’ . Materials and methods describing the review (which years ?, narrative or systematic ?) are absent.  The introduction starts with a mini-review of endometriosis.

Minor suggestions are

  • Do not overstate arguments eg L16-17 ‘ but retrograde menstruation and deficiency in natural killer (NK) cells to clear endometriotic cells in the peritoneal cavity play an important role.’ The role of NK cells is still hypothetic

It therefore is suggested to revise the paper with

  • an introduction introducing the subject without discussions as the clinical aspects of endometriosis and surgery, which are well known. Please limit endometriosis data to aspects that are important for this manuscript eg types of endometriosis ?
  • other aspects are missing or difficult to find such as:  are endometriotic cells similar to endometrium cells or immunological different ? an introduction to NK cells in the endometrium and in peritoneal fluid. Maybe these aspects could be structured in subheadings. At he end  the aim and rationale of the manuscript should be clear.
  • Materials and methods
  • Please structure reading in different immunological approaches and specify for which type of endometriosis if important.
  • Discussion should discuss the findings  eg why would endometrium and endometriosis cells react differently ? Why is the endometrium different in women with endometriosis , Why would NK therapy be specific for endometriosis ? etc

It will be a pleasure to read a revised, focused and structured manuscript

Author Response

Dear reviewer

Thank you for your comments on our manuscript. We revised the manuscript with your suggestions. We removed some parts of the introduction on clinical and surgical aspects of endometriosis, and tried to zoom in on aspects relevant for the further manuscript. In the result section we introduced subheadings and made a more structured overview.  Unfortunately it was not possible to answer all the questions suggested by the reviewer, because this information is not available (yet). We hope that more research will be preformed in teh field of endometriosis to answer all these interesting questions. 

We hope that these improvements on the manuscript, meet your requirements.

Reviewer 2 Report

The manuscript are well written and displayed. The authors reviewed  NK cell therapy as a new treatment options for endometriosis and listed the role of NK cells in the pathogenesis of endometriosis. However, one of the big issue that I should mention is that the  structure of this paper needed to be adjusted. Introduction/Results/Discussion /Material and Methods is too rough. It's not necessary to include that material parts. In the results parts can you separate and add in the subtitle of NK cell receptor, TGF beta or cytokines et ac which will make the whole paper more clean and clear.

Author Response

Thank you very much for your comments on our manuscript. We restructured the manuscript and introduced the subheadings as you suggested. We hope that this new version meets your requirements.

yours sincerely

Janneke Hoogstad

Round 2

Reviewer 1 Report

The readability if the manuscript “The Promises of natural killer cell therapy in endometriosis ‘’ has improved. It is a nice compilation of the changes in NK cells in endometriosis and of their potential therapeutic uses.

Minor suggestions are

  1. Add a short paragraph with the limitations of animal models in order to permit the median reader to grasp the message. Animal models might be endometrium in the peritoneal cavity. Anyway, if the growth and survival of these cells can be modulated, it is an important additional element of the role of immunology in the pathophysiology of endometriosis.
  2. State clearly that all data were obtained in animal models except when stated otherwise
  3. Add a short paragraph on the peritoneal cavity being a specific micro-environment for other many aspects such as steroid hormones and microbiota. Throughout the manuscript, it should be clear whether results are for PF or PB.  In the actual version, this is often confusing.
  4. The manuscript would be improved if it would be made clear that the effect described concerns the growth of these endometrium cells, not the initiation of endometriosis, whether seen as an invasion, or as genetic-epigenetic changes in the cell.

Author Response

Dear reviewer.

Thank you very much for the valuable suggestions. We addepted the manuscript and used all your suggestions to further improve the manuscript.

Concerning point 1 and 4 we added a part on the limitations of animal models and the limited possibilities of mimicking real endometriosis. You can find this part in line 242 to 245.

Concerning point 2, we added a sentence that all data were obtained in animal models except when stated otherwise. Line 147 to 148

In line 245 to 248 we added a few sentences on the specific microenvironment of the peritoneal cavity.

We hope that you think that this manuscript is suitable for publication in its current form. 

Thank you again for the valuable suggestions.

Yours Sincerely

Janneke Hoogstad-van Evert 
